

# Measurement report: Characteristics of nitrogen-containing organics in PM$_{2.5}$ in Urumqi, northwest China: differential impacts of combustion of fresh and old-age biomass materials

Yi-Jia Ma[1], Yu Xu[2,*], Ting Yang[1], Hong-Wei Xiao[2], Hua-Yun Xiao[2]

[1]School of Environmental Science and Engineering, Shanghai Jiao Tong University, Shanghai 200240, China

[2]School of Agriculture and Biology, Shanghai Jiao Tong University, Shanghai 200240, China

*Corresponding authors

Yu Xu

E-mail: xuyu360@sjtu.edu.cn



**Abstract:** Nitrogen-containing organic compounds (NOCs) are abundant and important aerosol components, deeply involving in global nitrogen cycle. However, the sources and formation processes of NOCs remain largely unknown, particularly in the city (Urumqi, China) farthest from the ocean worldwide. Here, NOCs in $PM_{2.5}$ collected in Urumqi over a one-year period were characterized by ultrahigh-resolution mass spectrometry. The abundance of CHON compounds (mainly poor-O unsaturated aliphatic-like species) in the positive ion mode was higher in the warm period than in the cold period, which was largely attributed to the contribution of fresh biomass material combustion (e.g., forest fires) associated with amidation of unsaturated fatty acids in the warm period, rather than the oxidation processes. However, CHON compounds (mainly nitro-aromatic species) in the negative ion mode increased significantly in the cold period, which was tightly related to old-age biomass combustion (e.g., dry straws) in wintertime Urumqi. For CHN compounds, we found that alkyl nitriles and aromatic CNH compounds showed higher abundance in the warm and cold periods, respectively. It further confirmed different impacts of the combustion of fresh- and old-age biomass materials on NOC compositions. Our results clarify the mechanisms by which fresh and old-age biomass materials emitted different NOCs.

**Keywords:** Aerosols, Organic nitrogen, Molecular composition, Fresh biomass, Old-age biomass

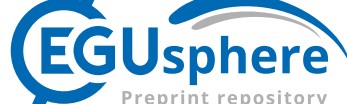

## 1. Introduction

Fine particulate matter ($PM_{2.5}$) is a typical atmospheric pollutant, which can affect
the global climate system, as well as urban air quality and human health (Seinfeld et al.,
2016; Wang et al., 2021a). Organic aerosol (OA) contributes significantly (20–90%) to
$PM_{2.5}$ mass concentration in most polluted areas worldwide (Zhang et al., 2007; Han et
al., 2023). However, up to 77% of molecules in OA include nitrogen-containing
functional groups (Ditto et al., 2020; Kenagy et al., 2021), which has been suggested to
play important roles in the formation, transformation, acidity, and hygroscopicity of OA
(Xu et al., 2020; Wang et al., 2017b; Laskin et al., 2009). Moreover, the modified forms
of some nitrogen-containing organic compounds (NOCs) and volatile organic
compounds (VOCs) by ozone ($O_3$), hydroxyl radical (•OH), and nitrogen oxide ($NO_x$)
can lead to an increase in the health hazards of OA, among which nitrated amino acids
and nitrated polycyclic aromatic hydrocarbons are two representative hazards (Franze
et al., 2005; Bandowe and Meusel, 2017). Thus, the identification of aerosol NOCs at
the molecular level is important for improving our understanding of the precursors,
sources, and formation processes of nitrogen-containing OA.
Previous observations in urban, rural, marine, and forest areas have suggested that
the molecular composition and relative abundance of aerosol NOCs were spatially
different (Samy and Hays, 2013; Jiang et al., 2022; Lin et al., 2012; Xu et al., 2023).
These differences can be mainly attributed to the diverse sources and formation
mechanisms of aerosol NOCs. Commonly reported primary sources include
combustion process releases and natural emissions (e.g., soils, plant debris, pollen, and



ocean) (Song et al., 2022; Wang et al., 2017b; Cape et al., 2011; Lin et al., 2023). In
addition, aerosol NOCs can also be tightly associated with secondary formation
processes involving the reactions of reactive nitrogen with VOCs or particle-phase
CHO compounds (Bandowe and Meusel, 2017; Zarzana et al., 2012; Laskin et al.,
2014). For example, laboratory experiments have suggested that the oxidation of
isoprene and α-/β-pinene in the presence of $NO_x$ can result in the formation of organic
nitrates (e.g., methacryloyl peroxynitrate, dihydroxynitrates, and monohydroxynitrates)
(Surratt et al., 2010; Rollins et al., 2012; Nguyen et al., 2015). The reduced nitrogen
species (e.g., $NH_3$, $NH_4^+$, and organic amines) have been demonstrated to contribute to
the formation of NOCs through "carbonyl-to-imine" transformations in the laboratory
experiments (Zarzana et al., 2012; Laskin et al., 2014). In the field observation studies,
NOCs in particulate matter were analyzed at the molecular level to indicate on their
sources and formation mechanisms (Jiang et al., 2022; Lin et al., 2012; Zhong et al.,
2023). Xu et al. (2023) characterized the variations of molecular compositions in urban
road $PM_{2.5}$, suggesting that organic nitrates increased largely through the interactions
of atmospheric oxidants, reactive gas-phase organics, and aerosol liquid water. Several
field studies conducted in Beijing (China) and Guangzhou (China) also suggested that
the molecular compositions and formation of NOCs were tightly associated with the
environmental conditions (Jiang et al., 2022; Lin et al., 2012; Xie et al., 2020).
Generally, most of studies on aerosol NOCs were performed in economically developed
regions, as well as in forest and marine areas (Jiang et al., 2022; Wang et al., 2017a;
Ditto et al., 2022b; Altieri et al., 2016; Miyazaki et al., 2014). In contrast, few studies





have investigated the sources and atmospheric transformation of NOCs in the northwest
border urban of China (e.g., Urumqi) with fragile ecology and harsh environmental
conditions (e.g., cold winter and dry summer), which may hinder our comprehensive
and in-depth understanding of the formation process of NOCs in ambient aerosols.
Biomass burning emissions were widely reported in the source identification of
aerosol NOCs in northern and southwestern China because of heating and cooking
needs (Zhong et al., 2023; Wang et al., 2021c; Chen et al., 2017). A recent observation
study in urban Tianjin suggested that most CHON compounds in wintertime $PM_{2.5}$
originated from biomass burning (Zhong et al., 2023). The $CHN_2$ compounds have been
identified in biomass burning OA (BBOA) (Laskin et al., 2009; Wang et al., 2017b).
Moreover, the high temperature generated by biomass burning can facilitate the release
of ammonia, a process which caused the reaction of carboxylic acids (e.g., oleic acid)
with ammonia to form amides and alkyl nitriles (Radzi Bin Abas et al., 2004; Simoneit
et al., 2003). Interestingly, we found that biomass burning in rural China typically
includes both fresh biomass materials (e.g., forest fires) and old-age biomass materials
(e.g., straw after autumn harvest, fallen leaf, and deadwood). Fresh biomass is rich in
oils and proteins, whereas old-age biomass materials are usually oligotrophic due to the
transfer of nutrients to tender tissues or fruits (Jian et al., 2016; Xu and Xiao, 2017).
Thus, NOCs released from different types of biomass combustion may vary in
molecular compositions. However, there are large gaps in our current knowledge about
the impacts of fresh and old-age biomass burning on NOCs in ambient aerosols.
Urumqi (northwest China) is the largest inland city farthest from the ocean in the



113 world, which is becoming increasingly prominent due to the national strategy of the

114 "One Belt, One Road". The city and neighboring countries have a dry summer that can

115 easily trigger forest fires (Bátori et al., 2018; Xu et al., 2021), while the winter is very

116 cold with intensive old-age biomass and fuel combustion for heating (Ren et al., 2017).

117 In this study, we presented one-year ambient measurements of the chemical

118 compositions in PM$_{2.5}$ collected from Urumqi. The specific aims of this study are (1) to

119 investigate the molecular-level speciation of functionalized organic nitrogen

120 compounds via a high-resolution mass spectrometry with positive (ESI+) and negative

121 (ESI−) ionizations and (2) to investigate the potential sources and formation processes

122 for NOCs with a special focus on the relative influences of fresh and old-age biomass

123 burning.

124

125 **2. Materials and methods**

126 **2.1. Study site description and sample collection**

127 The study was conducted in Urumqi city with an average altitude of 800 m. The

128 region has an arid temperate continental climate with an annual mean temperature of

129 7.4 ± 13.9 °C and an annual mean rainfall of 27.8 mm. The sampling site is located in

130 the suburban area (Boda campus of Xinjiang University) of the city (87.75°E, 43.86°N)

131 (**Figure S1**), which is characterized by low population and traffic density. This is

132 because Urumqi is relatively vast and sparsely populated compared to developed

133 coastal cities in China (Qizhi et al., 2016). Additionally, the area is surrounded by

134 mountains on three sides, resulting in the difficulty in diffusion of air pollutants. The





dominant forest trees in this area are *Picea schrenkiana*, *Betula tianschanica* Rupr.,
*Populus talassica* Kom., and *Ulmus pumila* L.. The dry climate and strong sunlight in
the warm period ($18.81 \pm 6.4$°C, **Table S1**) would be the main culprits of forest fires in
the local and nearby areas. In the cold period ($-1.96 \pm 11.26$°C) (**Table S1**), the
centralized heating and old-age biomass burning may be the main contributors of local
air pollution. Thus, it provides an unexpected opportunity to investigate the potentially
differential impacts of fresh and old-age biomass burning on aerosol NOCs.
A high-volume air sampler (Series 2031, Laoying, China) was set up on the
rooftop of a building (School of Geology and Mining Engineering, Xinjiang University).
PM$_{2.5}$ samples ($n = 73$) were collected every 5 days with a duration of ~24 h onto
prebaked (450 °C for ~ 10 h) quartz fiber filters (Pallflex, Pall Corporation, USA) from
1 March 2018 to 26 February 2019. One blank filter was collected every month ($n =$
12). All filter samples were stored at $-30$°C until further analysis. The meteorological
data (e.g., temperature and relative humidity) and the concentrations of O$_3$ and NO$_x$
were daily recorded from the adjacent environmental monitoring station during the
sampling campaigns. In addition, the trajectories (72 h) of air masses arriving at the
sampling site at each sampling event were calculated to investigate the potential
influence of pollutant transport on aerosol NOCs.

**2.2. Chemical analysis**
A portion of each filter sample was extracted twice using methanol (LC-MS grade,
CNW Technologies Ltd.) under sonication in a chilled ice slurry (~4 °C). The extracted



solutions were filtered through a polytetrafluoroethylene syringe filter (0.22 $\mu$m, CNW
Technologies GmbH). Subsequently, the extracts were concentrated to 300 $\mu$L with a
gentle stream of gaseous nitrogen (Shanghai Likang Gas Co., Ltd). The final extracts
were divided into two parts, which were analyzed separately as described in previous
study (Wang et al., 2021b) under ESI+ and ESI− modes using an UPLC-ESI-QToFMS
(Xevo G2-XS QToFMS, Waters) system. It should be pointed out that UPLC-ESI-MS
(i.e., TOF-only) was used to identify molecular formulas of organic matter, while the
functional groups of the target molecule formulas were deciphered by UPLC-ESI-
MS/MS (i.e., tandem mass spectrometry). Ions obtained from $m/z$ 50–700 were
assigned molecule formulas via assuming hydrogen or sodium adducts in ESI+ mode
and deprotonation in ESI− mode. Detailed chromatographic conditions, parameter
selection, and quality control were displayed in the Supplement (**Sect. S1**). Notably,
there may be differences in ionization efficiencies between compound types. However,
the exact impacts of ionization efficiency on multifunctional compounds in a complex
mixture are uncertain and difficult to evaluate (Ditto et al., 2022b; Yang et al., 2023).
Thus, the intercomparison across compound relative abundance without considering
potentially differentiated ionization efficiency was conducted in this study, which was
similar to many previous studies (Xu et al., 2023; Jiang et al., 2022).

For the measurement of inorganic ions, a portion of each filter sample was

ultrasonically extracted with Milli-Q water (18 M$\Omega$ cm) in an ice-water bath (~4 °C).
The extract solutions were then filtered via a polytetrafluoroethylene syringe filter (0.22
$\mu$m, Millipore, Billerica, MA). The concentrations of water-soluble inorganic ions





including $NO_3^-$, $SO_4^{2-}$, $Cl^-$, $Ca^{2+}$, $Mg^{2+}$, $Na^+$, and $NH_4^+$ in the samples were determined
using an ion chromatograph system (Dionex Aquion, Thermo Scientific, USA) (Xu et
al., 2022a; Lin et al., 2023).

**2.3. Compound categorization and predictions of ALW, pH, and hydroxyl radical.**

The molecular formulas identified by UPLC-ESI-QToFMS were classified into
several major compound classes based on their elemental compositions (i.e., C, H, O,
and N), primarily including CHO, CHON, and CHN groups in the ESI+ mode and CHO
and CHON groups in the ESI– mode (Wang et al., 2017b). All of the detected molecules
were reported as neutral molecules, unless stated otherwise. The double-bond
equivalent (DBE) and carbon oxidation state ($OS_C$) were calculated to reflect the
unsaturation degree of the organics and the composition evolution of organics that
underwent oxidation processes, respectively (details in **Sect. S2**) (Kroll et al., 2011; Xu
et al., 2023). Additionally, the modified aromaticity index ($AI_{mod}$) was also calculated
to indicate the aromaticity of organic compounds (details in **Sect. S2**) (Koch and
Dittmar, 2006).
A thermodynamic model (ISORROPIA-II) was applied to predict the mass
concentration of aerosol liquid water (ALW) and the value of pH with particle-phase
ion concentrations as well as ambient temperature and relative humidity as the inputs,
as detailed in our previous publications (Xu et al., 2020; Xu et al., 2023; Xu et al.,
2022b). The concentrations of ambient •OH were predicted using empirical formula
(Ehhalt and Rohrer, 2000; Wang et al., 2020).




## 3. Results and discussion

### 3.1. Overall molecular characterization of organic aerosols

**Figures 1a** and **1c** show the mass spectra of organic compounds detected in ESI+

and ESI−, respectively. More compounds were identified in ESI+ (1885 molecular

formulas) than in ESI− (438 molecular formulas) (**Table S2**), which was similar to

previous reports about the molecular characteristic of biomass burning aerosols and

urban aerosols (Jiang et al., 2022; Wang et al., 2017b). The molecular weights of the

compounds with relatively high signal intensity mainly ranged from 100 Da to 500 Da

in ESI+, which was larger than those (100–300 Da) observed in the urban (Changchun,

Guangzhou, and Shanghai) (Wang et al., 2021a) and agriculture (Suixi) (Wang et al.,

2017b) regions of China. In contrast, the species with the strong signal intensity fell

between 100 Da and 300 Da in ESI−. This mass range detected in Urumqi organic

aerosols was comparable to previous observations in urban aerosols (Han et al., 2023)

but significantly lower than that in firework-related urban aerosols (300–400 Da) (Xie

et al., 2020). On average, the molecular number and relative abundance of CHON

compounds (150–500 Da) were dominant in ESI+, accounting for 45.99% of the total

molecular number and 62.70 ± 6.83% of the total signal intensity (**Figures 1a** and

**Table S2**). CHO compounds were the second most abundant categories (28.76 ± 4.75%

of the total signal intensity), followed by CHN compounds. However, previous

observations conducted in Shanghai, Guangzhou, and Changchun suggested that the

compounds in ESI+ were dominated by CHN and CHON species (Wang et al., 2021a).



In ESI−, although the number of CHON compounds was less than CHO, the relative
abundance of CHON compounds (150–250 Da) was higher (**Figures 1d** and **Table**
**S2**). The finding was consistent with the results obtained in Shanghai and Changchun
but different from the case in Guangzhou (Wang et al., 2021a). The average H/C ratios
of CHO (1.62−1.66) and CHON (1.79−1.83) compounds in ESI+ mode (**Table S3**)
were higher than those (0.94−1.13 and 1.27−1.47) in Changchun, Shanghai, and
Guangzhou (Wang et al., 2021a). However, the average O/C ratios of CHO (0.25−0.3)
and CHON (0.22−0.3) compounds in ESI+ mode (**Table S3**) were less than those
(0.42−0.43 and 0.27−0.45) in the urban areas (Shanghai and Guangzhou) (Wang et al.,
2021a). Overall, these dissimilarities in molecular characteristics of organic aerosols
between Urumqi and other areas may be attributed to their different sources and
formation mechanisms.
**Figures 1b** and **1d** show the time series of the fractional distributions of various
organic matter categories in different ion modes. The abundance of CHO compounds
in ESI+ exhibited a temporal variation similar to that of CHON compounds ($r = 0.51$,
$P < 0.01$), with increased levels in the warm period. This indicated that CHO
compounds may be important precursors for the formation of NOCs or that they have
similar origins. Previous simulation experiment has demonstrated that higher
temperatures can result in an increase in the concentration of the oxygenated organic
molecules, while lower temperatures can allow less oxidized species to condense
(Stolzenburg et al., 2018; Frege et al., 2018). In addition, solar radiation and
atmospheric oxidation capacity are also important factors promoting the formation of



more oxygenated organic molecules (Li et al., 2022; Liu et al., 2022). Air temperature,
radiation, and atmospheric oxidation capacity were much higher in the warm period
than in the cold period in Urumqi (**Table S1**) (Wan et al., 2021), which may be partly
responsible for increased abundances of CHO and CHON compounds in the warm
period. However, the abundance of CHN compounds tended to increase from the warm
period to the cold period. Since the ESI+ mode is highly sensitive to protonatable
species, organic amines were expected to predominate the CHN compounds (Han et al.,
2023; Wang et al., 2021a). It is well documented that the formation of amine salt in the
particle phase is tightly associated with aerosol acidity and water (Liu et al., 2023).
Thus, the reduced pH value and increased ALW level in the cold period (**Table S1**)
provided greater potential for converting gaseous amines into particles.

In ESI− mode, the abundances of CHON and CHO exhibited a significantly

increased level in the cold period (**Figure 1d**), a variation pattern which was completely
opposite to the case in ESI+ mode. The ESI− mode is more sensitive to deprotonatable
compounds, such as nitrophenols, organic nitrates, organosulfates, and organic acids
(Jiang et al., 2022; Lin et al., 2012). The formations of these compounds were highly
impacted by ALW and aerosol acidity (Ma et al., 2021; Smith et al., 2014; Zhou et al.,
2023; Xu et al., 2023). However, Urumqi has dry and dusty weather, particularly in
warm period, resulting in a quite low ALW concentration ($1.86 \pm 1.90$ μg m$^{-3}$) in the
warm period (**Table S1**). Moreover, the calculated mean pH values were 6 during the
warm period without considering the influence of gaseous ammonia (**Table S1**).
Previous studies have suggested that a bias correction of 1 unit should be considered





for the prediction of aerosol pH when lacking of ammonia measurements (Guo et al.,
2015; Wang et al., 2021c). This implied that the actual aerosol acidity in the warm
period in Urumqi should be neutral or slightly alkaline. Obviously, the aerosol
characteristics of the warm period in Urumqi may hinder the formation of these organic
compounds measured in ESI− mode. In contrast, the increased ALW concentration and
decreased pH value during the cold period can facilitate the formation of CHO and
CHON compounds through the partitioning of gas-phase species to the particles and
subsequent aqueous phase reactions (Xu et al., 2020; Xu et al., 2023). Furthermore, the
total signal intensity of CHO compounds was significantly correlated with that of
CHON ($r = 0.62$, $P < 0.01$), indicating that they may have similar origins or that CHO
compounds may serve as important precursors for CHON compound formation. It
should be noted that this study mainly focuses on NOCs, therefore sulfur-containing
species were not discussed. In general, the differentiated seasonal variation patterns for
the different types of NOCs measured here can be attributed to the unique
meteorological conditions in Urumqi and different ionization mechanisms in ESI+ and
ESI− modes. The sources and formation mechanisms of NOCs will be further discussed
in the following sections.

**3.2. Seasonally differential sources and formation mechanisms of CHON**
**compounds**

CHON compounds can be products of reactions between CHO species and

reactive nitrogen species ($NO_x$, $NH_3$, and $NH_4^+$) (Lee et al., 2016; De Haan et al., 2017),



as also partly implied by significant positive correlations ($r = 0.51–0.62$, $P < 0.01$)
between total signal intensity of CHO and CHON compounds in both ESI+ and ESI−
modes. Thus, CHO compounds were further classified based on their $OS_C$ values to
preliminarily explore their origins and linkages with CHON compound formation
(**Figures 2a** and **2b**). In ESI+ mode, the $OS_C$ values of the detected CHO compounds
(−1.75 to 0.5) were higher than those of primary vehicle exhausts (−2.0 to −1.9) (Aiken
et al., 2008), likely indicating a weak (or indirect) contribution of primary vehicle
exhausts to CHO molecules in Urumqi. The signal intensity of BBOA dominated the
total OA signal intensity and was higher in the warm period than in the cold period
(**Figure 2e**). However, previous studies conducted in China (e.g., Beijing, Xi'an,
Shanghai, and Liaocheng) suggested that biomass burning was more significant in the
cold seasons (Li et al., 2023; Wang et al., 2017a; Chen et al., 2017; Wang et al., 2009;
Wang et al., 2018). Furthermore, we found that the oxygen-poor unsaturated aliphatic
compounds showed a high signal intensity in the warm period and that the signal
intensities of all categories of compounds in the warm period were weakly correlated
with atmospheric oxidants (i.e., $O_3$ and •OH) ($r < 0.1$, $P > 0.05$). Thus, the formation or
source of CHO compounds in the warm period may not be mainly controlled by high
atmospheric oxidation, but rather by biomass burning, which was distinguished from
previous reports (Duan et al., 2020; Kondo et al., 2007). This consideration was also
supported by the fact that there were significantly more fire spots in the warm period
than in the cold period (**Figure 3**). It should be noted that the materials used for biomass
burning in the cold period in rural China are typically old-age plant tissues (**Figure S3**),

while biomass burning in the warm season is mainly attributed to forest fires or wildfires (relatively fresh biomass). Accordingly, a large number of fresh biomass material burning occurred from April to October each year in the neighboring countries (e.g., Kazakhstan) (Xu et al., 2021) or regions of Urumqi (due to drought) (**Figure 3**) may be largely responsible for high CHO compound abundance in the warm period.

The CHO species in ESI− had higher $OS_C$ (–1.85 to 1.1) than those in ESI+ (**Figures 2c** and **2d**), which was consistent with a recent study conducted in Guangzhou, China (Zou et al., 2023). The predominant subgroups of CHO in ESI− were BBOA and semivolatile oxidized OA (SV-OOA), which was different from the observation in Shanghai (dominated by SV-OOA and low-volatility oxidized OA) (Wang et al., 2017a). Additionally, some specific saturated and unsaturated aliphatic CHO substances (i.e., $C_{12–18}H_nO_2$) in ESI− showed higher abundance in the warm season than in the cold season, which was contrary to the variation pattern of other CHO compounds. These $C_{12–18}H_nO_2$ compounds were found to be mainly fatty acids, such as stearic acid ($C_{18}H_{36}O_2$), oleic acid ($C_{18}H_{34}O_2$), linolelaidic acid ($C_{18}H_{32}O_2$), palmitic acid ($C_{16}H_{32}O_2$), and palmitoleic acid ($C_{16}H_{30}O_2$) (**Figure S4**), all of which usually accumulate in plants, particularly *Suaeda aralocaspica* (W. Hogg and T. Gillan, 1984; Wang et al., 2011). Interestingly, this plant was widely distributed in Central Asia as well as in the southern edge of the Junggar Basin in Xinjiang, China (Wang et al., 2011). Although fatty acids can also originate from food cooking (Zhao et al., 2007), there seems to be no seasonal differences in cooking behavior locally. Thus, these results further confirmed our consideration that the abundance of CHO compounds in the



warm period was highly impacted by fresh biomass material burning (e.g., forest fires
or wildfires).

CHON molecules in ESI+ were mainly identified as unsaturated aliphatic-like

compounds with poor oxygen (**Figures 4a** and **4b**), accounting for more than 70% of
the total signal intensities of CHON species (**Figure S5**). The signal intensity of CHON
species in ESI+ was greater in the warm period than in the cold period (**Figure 4e**).
Moreover, BBOA contributed to 56.9 % of the total CHON signal intensity in the warm
period (**Figure S6**). These characteristics of CHON compounds were similar to those
of CHO. Considering a significant positive correlation ($r = 0.62$, $P < 0.01$) between the
total signal intensity of CHO and CHON compounds in ESI+, we thus concluded that
primary sources (i.e., fresh biomass material burning) were also one of the main sources
of CHON compounds. In this study, CHON compounds with O/N < 3 contributed 76.48
± 1.11% of total CHON species in ESI+ (**Figure S7**), which was much larger than the
results observed in urban Tianjin in winter (less than 20%) (Zhong et al., 2023). In
particular, $C_{16}H_{33}ON$, $C_{18}H_{37}ON$, $C_{18}H_{35}ON$, $C_{18}H_{33}ON$, $C_{18}H_{31}ON$, and $C_{20}H_{33}ON$
showed a high abundance, together accounting for 55.04 ± 7.09 % of the total CHON
abundance (**Table S4**). The carbon number of these compounds was consistent with
that of fatty acids mentioned above; moreover, their abundances showed a positive
correlation ($r = 0.43–0.81$, $P < 0.01$) with the abundances of corresponding fatty acids
in the warm period. In contrast, these CHON compounds only showed a weak
correlation ($r = -0.24 \sim 0.33$) with atmospheric oxidants (e.g., •OH, $O_3$, and $NO_x$). Thus,
the formation mechanism of biomass burning-related NOCs in Urumqi during the warm



period may be the interaction between fatty acids and reduced nitrogen species (e.g.,
$NH_3$) rather than the oxidation pathway involving CHO compounds and $NO_x$.
A recent laboratory study has suggested that $NH_3$ produced during the thermal
degradation of amino acids can react with oleic acid from the pyrolysis of triglycerides
to form amides (R1) (Ditto et al., 2022a). As discussed above, the combustion of fresh
biomass materials (e.g., forest fires or wildfires) can release abundant fatty acids. In
addition, wildfires can also emit large amounts of $NH_3$, with an average emission factor
more than twice $NH_3$ emission factor of agricultural fires (Tomsche et al., 2023).
According to MS/MS analysis (**Table S5**), potential fatty acid-derived NOCs were
indeed identified as amides. Thus, we proposed that the high temperature generated
during wildfires or forest fires provides suitable conditions for the reaction of
carboxylic acids and $NH_3$ to from amides. The specific process was presented in **Figure**
**5** (Pathway 1). It has been suggested that atmospheric oxidants can oxidize olefins (R2
and R3) to form hydroxyl nitrates and carbonyl nitrates (Perring et al., 2013). Therefore,
fatty acids (oleic acid as a representative) released from fresh biomass material burning
may also rely on oxidation pathways to form NOCs (**Figure 5**, Pathway 2). It is worth
noting that some products with double bonds after the amidation of unsaturated fatty
acids can continue to undergo the reactions of R2 and R3 in the atmosphere, resulting
in the formation of nitrooxy amides (**Figure 5**, Pathway 3). However, we found that the
abundance of oleic acid-derived amides via Pathway 1 in the warm period was more
than 100 times higher than that of NOCs with $-ONH_2$ (thus, the impact of ionization
efficiency is expected to be less than 100 times) from Pathways 3. In the cold period,





the abundance of fatty acids-derived amides decreased dramatically (**Figure 5** and
**Figure S8**). Thus, the overall results demonstrated that the combustion of fresh biomass
materials indeed contributed significantly to aerosol NOCs (e.g., amides) in the warm
period in Urumqi.
$RCOOH \xrightarrow{NH_3, -H_2O, \text{High temperature}} RCONH_2$                                     (R1)
$RH \xrightarrow{\bullet OH} R\bullet \xrightarrow{O_2} RO_2\bullet \xrightarrow{NO} RONO_2$                                     (R2)
$R_1=R_2 \xrightarrow{NO_3\bullet} R_1(ONO_2)\text{-}R_2\bullet \xrightarrow{O_2} R_1(ONO_2)\text{-}R_2O_2\bullet \xrightarrow{RO_2, NO_3} R_1(ONO_2)\text{-}R_2(O)$     (R3)
The CHON species detected in ESI− were mainly aromatic-like compounds,
whose signal intensities were significantly greater in the cold period than in the warm
period (**Figures 4c,4e** and **Figure S5**). Moreover, we found that several nitro-aromatic
compounds, including $C_6H_5O_3N$, $C_6H_5O_4N$, $C_7H_7O_3N$, $C_7H_7O_4N$, $C_7H_5O_5N$, $C_8H_9O_3N$
(confirmed by their authentic standards in the LC/MS analysis), contributed up to 50%
of the total CHON (ESI− mode) intensity (**Table S6**). Other NOCs with relatively high
signal intensity were mainly $O_{4\text{-}6}N_2$ species (contributed up to 25%), such as $C_6H_4O_5N_2$,
$C_7H_4O_7N_2$, $C_7H_6O_5N_2$, and $C_7H_6O_6N_2$, which have been suggested to be associated with
secondary photochemical or multiphase chemical processes (Harrison et al., 2005;
Cecinato et al., 2005; Salvador et al., 2021). However, the abovementioned nitro-
aromatic compounds including $C_6H_5O_3N$ (nitrophenol), $C_6H_5O_4N$ (nitrocatechol),
$C_7H_7O_3N$ (methyl-nitrophenol), $C_7H_7O_4N$ (methyl-nitrocatechol) were primarily
identified as tracers of straw and wood burning (old-age biomass materials commonly
used in suburban and rural China) (Iinuma et al., 2010; Kourtchev et al., 2016). A study



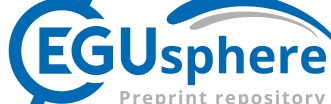

about molecular characterization (ESI− mode) of water-soluble aerosols emitted from
the combustion of old-age biomass materials (i.e., dry corn straw, rice straw, and pine
branches) and coal showed that OA from old-age biomass burning typically contained
much more nitro compounds and/or organonitrates than that from coal, while OA from
coal-smoke contained more sulfur-containing compounds (Song et al., 2018). Thus, the
old-age biomass burning associated with winter heating rather than coal combustion
may contribute a significant amount of aerosol NOCs (e.g., nitrophenols) in wintertime
Urumqi. However, it does not necessarily suggest that the importance of multiphase
chemistry in the formation of NOCs was ignorable, as indicated by relatively high
signal intensity of $O_{4-6}N_2$ species. In general, the differential molecular characteristics
of CHON species in different seasons in Urumqi can largely attributed to different
impacts of the combustion of fresh- and old-age biomass materials.

**3.3. CHN Molecule Evidence of Fresh and Old-age Biomass Burning in Different Periods.**

**Figures 6a** and **6b** present the van Krevelen diagram of CHN compounds in the
cold and warm periods. The $CHN_1$ compounds with relatively high signal intensity
mainly contained 7–20 carbon atoms, among which $C_5H_5N(CH_2)_n$, $C_9H_7N(CH_2)_n$, and
$C_{13}H_9N(CH_2)_n$ were dominant (78.68 ± 7.59 % of the total signal intensity of $CHN_1$
compounds in the cold period, **Table S7**). $C_5H_5N(CH_2)_n$ could be identified as pyridine
and its homologues, which have been detected in freshly discharged BBOA (Dou et al.,
2015). Additionally, the abundance of $C_5H_5N(CH_2)_n$ was positively correlated with that





of $C_9H_7N(CH_2)_n$, $C_{13}H_9N(CH_2)_n$, and nitro-aromatic compounds mentioned above ($r =$
$0.46-0.81$, $P < 0.01$), particularly in the cold period with old-age biomass burning for
heating. We further found that both the total signal intensity and aromaticity of $CHN_1$
species was much higher in the cold period ($AI_{mod}$ of 0.52) than in the warm period
($AI_{mod}$ of 0.35) (**Figure 6** and **Figure S9**). It has been suggested that old-age leaves
contain more aromatic compounds compared to fresh leaves (Jian et al., 2016). Thus,
the overall results implied that old-age biomass burning had an important contribution
to the variation of $CHN_1$ compounds. In particular, the intensity of $CHN_1$ compounds
was significantly negatively correlated with the concentration of $O_3$ and $\cdot OH$ ($r = -0.44$
$\sim -0.53$, $P < 0.01$), suggesting that atmospheric oxidation processes were the potential
pathway for amine removal rather than the sources of particle amine salts (Zahardis et
al., 2008; Qiu and Zhang, 2013). This result was different from the previous case
showing the formation processes of $CHN_1$ and its homologs in Guangzhou (South China)
were tightly related to photo-oxidation processes (Jiang et al., 2022). The $CHN_2$ species
showed a similar temporal variation pattern to the $CHN_1$ species. Moreover, the
abundances of total $CHN_2$ and major components ($C_{8-11}H_8N_2(CH_2)_n$, $C_{10}H_{14}N_2(CH_2)_n$,
$C_{10}H_{16}N_2(CH_2)_n$ and $C_5H_8N_2(CH_2)_n$) were positively correlated with that of total $CHN_1$
($r = 0.55-0.90$, $P < 0.01$), but negatively correlated with the concentration of $O_3$
and $\cdot OH$ ($r = -0.43 \sim -0.60$, $P < 0.01$). Clearly, old-age biomass burning, particularly
in the cold period, also exerted significant impacts on the abundance of $CHN_2$
compounds, which was also supported by several previous studies (Laskin et al., 2009;
Wang et al., 2017b; Song et al., 2022).



Interestingly, we found some CHN species with 16–20 carbon atoms showed
higher abundance in the warm period than in the cold period, a pattern of which was
opposite to that of all other CNH compounds (**Figure 6c**). These $C_{16-20}N_1H_x$
compounds were further identified as alkyl nitriles (**Table S5**) (Simoneit et al., 2003).
In addition, the carbon number of the identified alkyl nitriles was consistent with those
of amides previously proposed to be produced by fresh biomass burning. Thus, we
proposed that fresh biomass material burning in the warm period may provide a
continuous high-temperature environment to promote the dehydration of amides
(**Figure 5**, Pathway 4). These alkyl nitriles with double bonds can continue to undergo
the reactions of R2 and R3 (**Figure 5**, Pathway 5). However, the signal intensity of the
nitrooxy products in the warm period was insignificantly correlated with the
concentration of $O_3$, ·OH, and $NO_x$ ($P > 0.05$), likely indicating a weak influence of
atmospheric oxidation on alkyl nitrile removal in this site. The high-temperature
dehydration of amides (e.g., erucamide) to form alkyl nitriles (e.g., erucyl nitrile) has
been demonstrated by Simoneit et al. (Simoneit et al., 2003) in a laboratory simulation
experiment. A study on BBOA also showed that alkyl nitriles can be serve as indicators
of biomass burning in the ambient atmosphere (Radzi Bin Abas et al., 2004).
Furthermore, the abundance of identified alkyl nitriles initially increased from March
and peaked in September and October (**Figure S10**), a pattern of which was consistent
with the interannual variation in wildfire areas (more in the warm period) in Central
Asian countries (Xu et al., 2021). Although cooking is also a potential source of alkyl
nitriles (Schauer et al., 1999), this activity does not have seasonal differences. In





contrast, the dramatically increased abundance of aromatic CNH compounds in the cold
period (**Figure S9**) can be attributed to the aqueous reactions of amines emitted from
old-age biomass material and coal combustion with acidic substances, as indicated by
significant correlations ($r = 0.61-0.95$, $P < 0.01$) between total CHN abundance and
$SO_4^{2-}$ and $NO_3^-$ concentrations. These findings further confirmed that the NOCs from
the combustion of fresh biomass materials in the warm period in suburban Urumqi were
compositionally different from those from old-age biomass burning in the cold period.

**4 Conclusions**

The complexity of NOCs restricts our understanding of its sources and formation

processes. In this study, the molecular compositions of organic aerosols in $PM_{2.5}$
collected in Urumqi over a one-year period were systematically characterized in both
ESI− and ESI+ modes, with a major focus on NOCs. A large amount of NOCs were
identified, showing that NOCs in relatively highly oxidative and reduced forms can be
roughly distinguished via these two ionization modes. Based on the identification of
molecular markers of amides and alkyl nitriles (much higher in the warm period) and
the analysis of their formation mechanisms (less contribution of atmospheric oxidation),
we highlighted the important contribution of combustion of fresh biomass materials
such as forest fires and wildfires to NOCs in the warm season in Urumqi. In contrast,
the dramatically increased abundances of aromatic CNH compounds and nitro-aromatic
CHON compounds (mainly nitrophenols) in the cold period were tightly associated
with the impacts of old-age biomass material burning. These results were illustrated in



a diagram (**Figure 7**).

Biomass materials in rural China were typically old-age plant tissues, as

mentioned above. Fresh biomass materials (e.g., green vegetation) with the enrichment
of oils and proteins can exist in forest fires or wildfires. Indeed, previous studies have
suggested that biomass burning can lead to the formation of aerosol amines and nitriles.
However, no field observation studies have paid attention to the differences in aerosol
NOCs emitted from the combustion of fresh and old-age biomass materials. For the
first time, our results reveal that fresh biomass material combustion can contribute more
amines and nitriles than old-age biomass material combustion. Generally, this study
provides the field evidence on the differential impacts of combustion of fresh and old-
age biomass materials on aerosol NOCs, improving our current understanding of the
molecular compositions of organic nitrogen aerosols in a vast territory with a sparse
population in Northwest China. Moreover, according to the fact that the studied site is
highly affected by combustion emissions of different types of biomass materials, future
work is needed to deeply understand the quantitative contributions of different types of
biomass burning to OA in China.

**Data availability.** The data in this study are available at
https://doi.org/10.5281/zenodo.10453929

**Competing interests.** The authors declare no conflicts of interest relevant to this study.



**Supplement.** Details of chemical analysis and data processing, eight tables (Tables
S1−S8), and ten extensive figures (Figures S1−S10).

**Author contributions.** YX designed the study. YJM, TY, and HWX performed field
measurements and sample collection; YJM and TY performed chemical analysis; YX
and YJM performed data analysis; YX and YJM wrote the original manuscript; and YX,
YJM, HWX, and HYX reviewed and edited the manuscript.

**Financial support.** This study was kindly supported by the National Natural Science
Foundation of China through grant 42303081 (Y. Xu) and Shanghai "Science and
Technology Innovation Action Plan" Shanghai Sailing Program through grant
22YF1418700 (Y. Xu).

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





**Figure 1.**

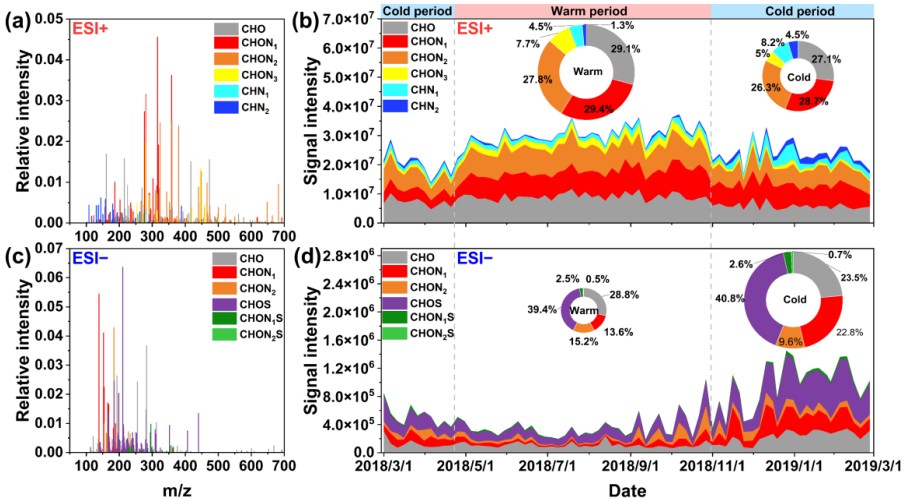

**Figure 1.** The reconstructed mass spectrum distribution of the detected species in PM$_{2.5}$

in (**a**) ESI+ and (**c**) ESI− modes during the whole campaign. Temporal variations in the

fractional distribution of classified compounds in (**b**) ESI+ and (**d**) ESI− modes. The

ring diagrams inside the panel show the signal intensity fractions of classified

compounds, the size of which is proportional to the total signal intensity of all species

detected in PM$_{2.5}$ in different periods.





**Figure 2.**

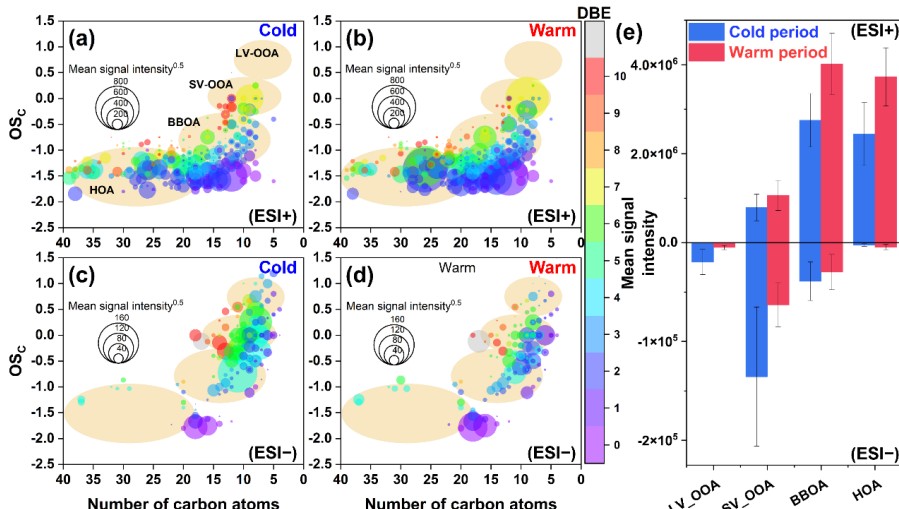


**Figure 2.** OSc values of CHO molecules detected in (**a** and **b**) ESI+ and (**c** and **d**) ESI−
modes in PM$_{2.5}$ collected from different periods (cold vs. warm). The size and color of
the circle indicate the mean signal intensity and DBE value of compounds, respectively.
The light-orange background indicates the areas of low-volatility oxidized OA (LV-
OOA), semivolatile oxidized OA (SV-OOA), biomass burning-like OA (BBOA), and
hydrocarbon-like OA (HOA) (Kroll et al., 2011), according to which (**e**) the mean signal
intensity of classified compounds was calculated for samples from different periods.








**Figure 3**

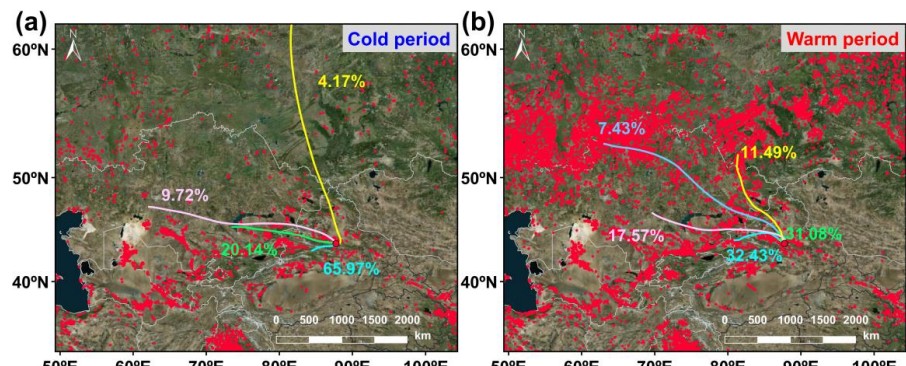

**Figure 3**. The 3-day (72 h) back trajectories illustrating the typical air mass flows to the study site during the (a) warm and (b) cool periods. Fire spots were shown in red, which was created based on NASA active fire data (VIIRS 375 m, https://firms.modaps.eosdis.nasa.gov/active_fire/). The map was derived from ©MeteoInfoMap (version 3.6.2) (Chinese Academy of Meteorological Sciences, China).

**Figure 4.**

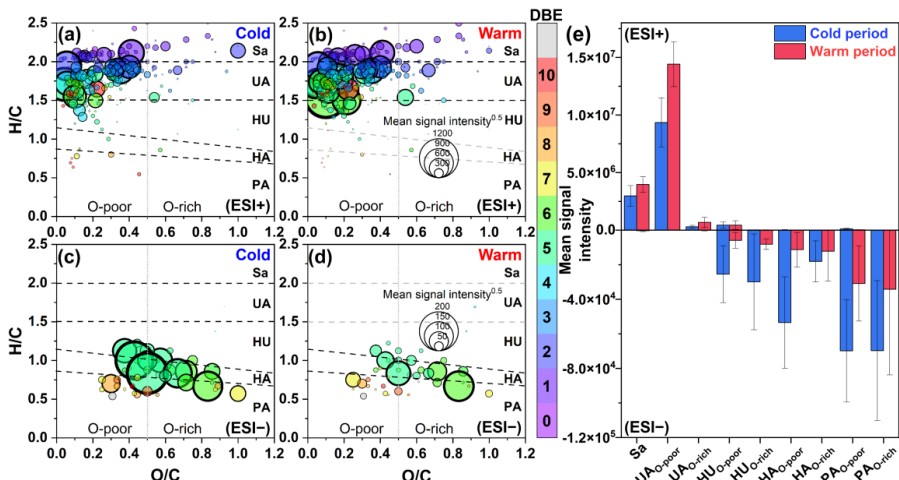

**Figure 4.** Van Krevelen diagrams of CHON molecules detected in (**a** and **b**) ESI+ and (**c** and **d**) ESI− modes in $PM_{2.5}$ collected from different periods (cold vs. warm). The subgroups in the panel include saturated-like (Sa), unsaturated aliphatic-like (UA), highly unsaturated-like (HU), highly aromatic-like (HA), and polycyclic aromatic-like (PA) compounds, further distinguishing between oxygen-poor and oxygen-rich compounds with an oxygen to carbon ratio of 0.5. The size and color of the circle indicate the mean signal intensity and DBE value of compounds, respectively. The (**e**) mean signal intensity of classified compounds was calculated for samples from different periods.



**Figure 5.**

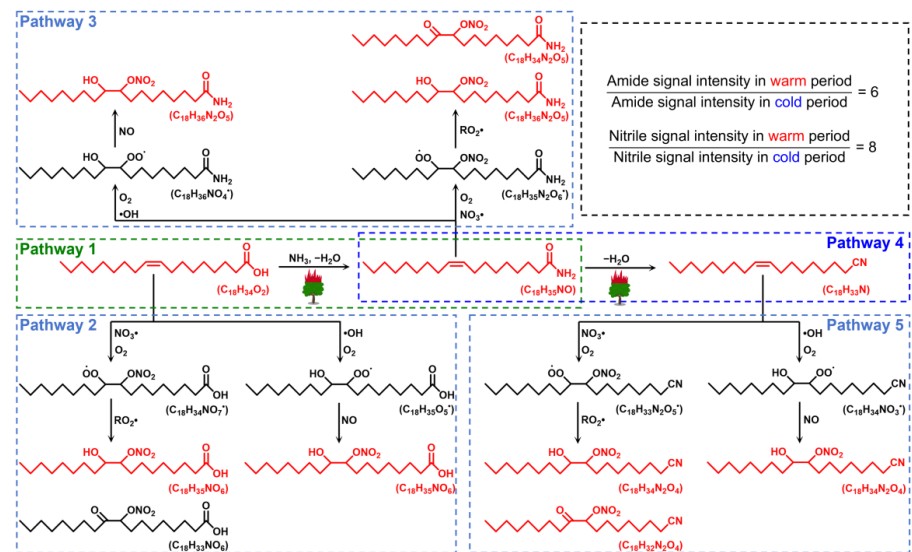


**Figure 5.** Proposed pathways for the reaction of carboxylic acids (oleic acid as a
representative) with ammonia to form the observed NOCs in $PM_{2.5}$ under the influence
of the high temperature generated during wildfires or forest fires. Compounds observed
in $PM_{2.5}$ were shown in red.





**Figure 6.**

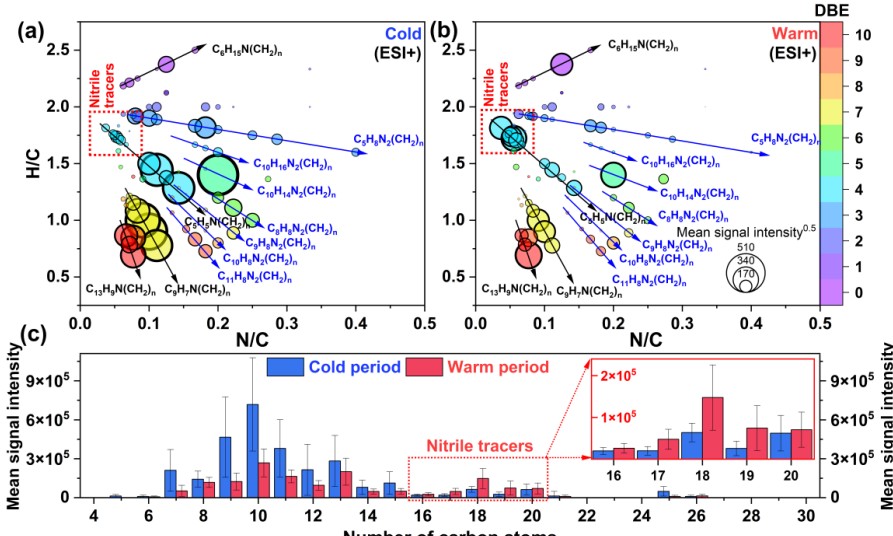


**Figure 6.** Van Krevelen diagrams of CHN molecules detected in PM$_{2.5}$ collected from
the (**a**) cold and (**b**) warm periods. The size and color of the circle indicate the mean
signal intensity and DBE value of compounds, respectively. The mean signal intensity
distributions of (**c**) carbon atoms in CHN molecules detected in PM$_{2.5}$ collected from
the cold and warm periods



**Figure 7.**

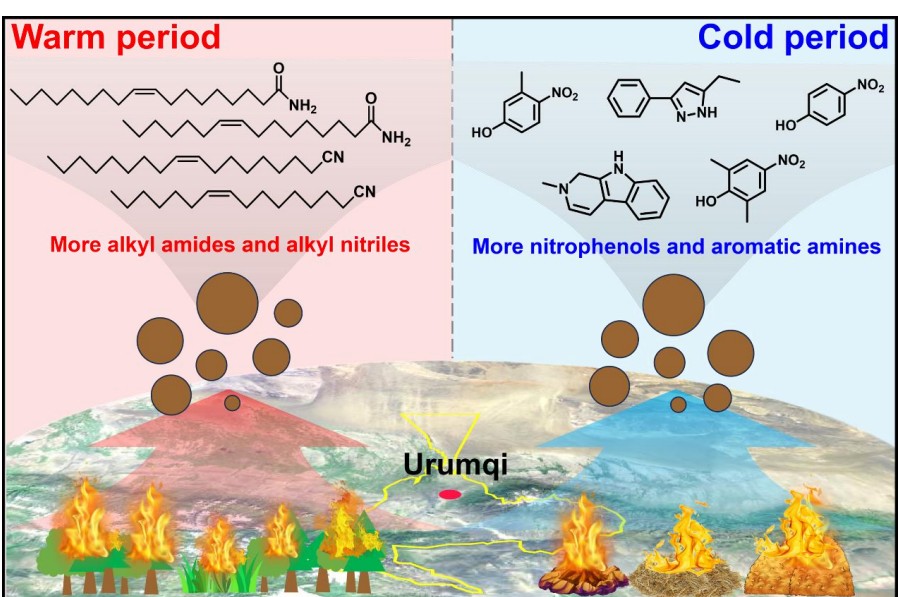

**Figure 7.** Conceptual picture showing the differential impacts of combustion of fresh and old-age biomass materials on aerosol NOCs in suburban Urumqi. The map was derived from ©Baidu Maps (BIDU, China).