# Peer review of "Measurement report: Characteristics of"

_EGUsphere, 2023_

## Author Comment (AC1)

**General.**

We would like to appreciate the editor and reviewers for providing the valuable comments and a better perspective on our work to improve the manuscript. In particular, we are very grateful to the editor and reviewers for giving us the opportunity to make revision. We have revised our manuscript by fully taking the reviewers' comments into account. Responses to specific comments raised by the reviewers are described below. All the changes made and appeared in the revised text are shown in red. All detailed answers to comments are displayed in blue.

**Comments of Referee #1 and our responses to them**

Comments:

*This study focused on the measurement and characterization of nitrogen-containing organic compounds in PM$_{2.5}$ collected in Urumqi over a one-year period. As mentioned in the manuscript, Urumqi is the largest inland city farthest from the ocean in the world. However, I have found that work on organic aerosols is rarely reported here. Thus, the manuscript can contribute a significant amount of valuable field data and will appeal to the readership in the field of atmospheric chemistry. Moreover, the authors present an interesting result indicating significant differences in the composition of aerosol nitrogen-containing organic compounds released from the combustion of fresh and old biomass materials. Biomass burning is usually a general concept in many previous studies, for which further refinement or classification is necessary. Thus, the topic is very meaningful. Overall, I recommend this paper for publication after addressing the following minor*

*comments.*

Response: We appreciate your professional review for our article. We have revised the manuscript to address the comments. Our responses to the specific comments and changes made in the manuscript are given below.

Specific comments:

1) *Lines 34–40: The expression is too concise, which may make it difficult for readers to understand why "It further confirmed different impacts of the combustion of fresh- and old-age biomass materials on NOC compositions". Please clarify it.*

Response: This section has been revised to enhance reader comprehension. See below for details (Lines 37–43).

Lines 37–43: …For CHN compounds, alkyl nitriles and aromatic species showed higher abundance in the warm and cold periods, respectively. Alkyl nitriles can from fresh biomass material combustion associated with the dehydration of amides (the main CHON compounds in the warm period). In contrast, aromatic species were tightly related to old-age biomass burning. These findings further suggested different impacts of the combustion of fresh- and old-age biomass materials on NOC compositions in different seasons…

2) *Some references about NOC should be cited in line 63-94. The Roles of N, S, and O in molecular absorption features of brown carbon in PM2.5 in a typical semi-arid megacity in northwestern China. Journal of Geophysical Research-Atmospheres, 2021, 126. Connecting oxidative potential with organic carbon molecule composition and source-specific apportionment in PM2.5 in Xi'an, China. Atmospheric Environment, 2023, 306, 119808.*

Response: We appreciate the introduction of these excellent and interesting references. All references you mentioned above have been added in the revised manuscript (Lines 66-67).

Lines 66-67: …(Samy and Hays, 2013; Jiang et al., 2022; Lin et al., 2012; Xu et al., 2023; Luo et al., 2023; Zeng et al., 2021)…

3) *Line 80: please delete "on"*

Response: The revision has been made in the revised manuscript (Line 82).

4) *Lines 266–269: This content involves the uncertainty of pH prediction. Thus, I suggest the author move this discussion to section 2.3. Compound categorization and predictions of ALW, pH, and hydroxyl radical. Furthermore, please clarify how pH is*

*predicted.*

Response: We thank you for these insightful comments. Based on your suggestion, we have modified this section as shown below (Lines 277–279).

Lines 277–279: …Moreover, the calculated mean pH value was 6.86 ± 1.71 (**Table S1**) during the warm period, which implies that the fine aerosol particles in the warm period in Urumqi was neutral or slightly alkaline.

Furthermore, we have clarified how pH is predicted in Section 2.3 Compound categorization and predictions of ALW, pH, and hydroxyl radical. The added descriptions in the revised manuscript are shown below (Lines 202–209).

Lines 202–209: …The model output results based on our data set showed that 94% and 90% of $NO_3^-$ were in the aerosol phase in the cold and warm periods, respectively. Hence, the predictions of pH and ALW were conducted without considering gaseous nitric acid (Guo et al., 2015; Wang et al., 2021). 78% and 21% of $NH_4^+$ were in the aerosol phase in the cold and warm periods, respectively. Moreover, it is important to note that gaseous $NH_3$ measurements were not conducted and ammonia partitioning was not considered in this study. Thus, a bias correction of 1 pH unit was applied to calculate the aerosol pH values (Guo et al., 2015; Wang et al., 2021).

5) *Lines 214–216: … urban aerosols… Please clarify the research site.*

Response: The revisions have been made in the revised manuscript (Lines 225–226).

6) *Lines 228 and 231: …94−1.13 for CHO and 1.27−1.47 for CHON…0.42–0.43 for CHO and 0.27–0.45 for CHON…*

Response: The revisions have been made in the revised manuscript (Lines 239–243).

7) *In section 3.2 Some references about sources profile should be cited to discussed the sources and formation mechanisms of NOC*

*Source profiles of molecular structure and light absorption of PM2.5 brown carbon from residential coal combustion emission in Northwestern China. Environmental Pollution, 2022, 299, 118866.*

*Optical properties, molecular characterizations, and oxidative potentials of different polarity levels of water-soluble organic matters in winter PM2.5 in six China's megacities. Science of The Total Environment, 2022, 853, 158600.*

*Insight into the Primary and Secondary Particle-Bound Methoxyphenols and Nitroaromatic Compound Emissions from Solid Fuel Combustion and the Updated Source Tracers. Environmental Science & Technology, 2023,57, 14280−14288.*

Response: We greatly appreciate your suggestions. All references you mentioned above have been added in the revised manuscript.

Lines 66–67: …(Samy and Hays, 2013; Jiang et al., 2022; Lin et al., 2012; Xu et al.,

2023; Luo et al., 2023; Zeng et al., 2021; Zhang et al., 2022; Zeng et al., 2020)…

Lines 91–92: …(Jiang et al., 2022; Wang et al., 2017; Ditto et al., 2022; Altieri et al., 2016; Xu et al., 2020; Liu et al., 2023; Zhang et al., 2022; Zeng et al., 2020)…

Lines 317–318: …(Duan et al., 2020; Kondo et al., 2007; Zhang et al., 2023)...

Lines 310-311: …(Li et al., 2023; Wang et al., 2017; Chen et al., 2017; Wang et al., 2009; Wang et al., 2018; Zhang et al., 2022).

8) *Lines 287–288: CHON compounds can be derived from the reactions between CHO species and reactive nitrogen species.*

Response: The revision has been made in the revised manuscript (Lines 297–298).

9) *Line 310: What are the main types of old-age plant tissues? Please clarify it.*

Response: We have clarified the main types of old-age plant tissues as described below (Lines 319–322).

Lines 319–322: …It should be noted that the materials used for biomass burning in the cold period in rural China are typically old-age plant tissues, such as dead branches of pine trees, dead branches of shrubs, corn straw, and rice straw (**Figure S3**), …

10) *Line 316: Please also provide the OSc range of CHO compounds in ESI+.*

Response: We thank you for the insightful comment. The $OS_C$ ranges of CHO compounds in ESI+ have been added in the revised manuscript (Lines 328–329).

11) *Lines 318–319: I suggest the authors provide percentage data for BBOA and SV-OOA.*

Response: The percentage data for BBOA and SV-OOA have been added in the revised manuscript (Lines 330–332).

12) *Lines 387–388 and 395: …C7H5O5N, and C8H9O3N (confirmed by their authentic standards), together contributed…C7H7O3N (methyl-nitrophenol), and C7H7O4N (methyl-nitrocatechol)…*

Response: The revision has been made in the revised manuscript (Lines 398–400 and 407–408).

13) *Line 456: Please change Simoneit et al. (Simoneit et al., 2003) to Simoneit et al. (2003).*

Response: The revision has been made in the revised manuscript (Line 480).

14) *Lines 424–427: Aromatic compounds can also originate from fossil fuel combustion in the winter period, please consider this possibility.*

Response: The added descriptions in the revised manuscript are shown below (Lines 454–466).

Lines 454–466: …A study about molecular characterization (ESI+ mode) of humic-like substances emitted from the combustion of old-age biomass materials (i.e., dry corn straw, rice straw, and pine branches) and coals showed that OA from old-age biomass burning typically contained much more $CHN_2$ compounds (55–64%) than that from coal (20–37%), while OA from coal-smoke showed more $CHN_1$ compounds (78–84%) compared to that from old-age biomass materials (15–22%) (Song et al., 2022). In this study, the signal intensity of $CHN_1$ compounds in the cold period was about 40% higher than that in the warm period, while that of $CHN_2$ compounds showed a 160% increase from the warm period to the cold period. Thus, although the contribution of fossil fuel (e.g., coal) combustion to NOCs in the cold period cannot be ignored, our results at least suggested that the biomass burning-derived CHN compounds showed a more significant increase compared to coal combustion-derived compounds from the warm period to the cold period in Urumqi.

**At last, we deeply appreciate the time and effort you've spent in reviewing our manuscript.**

**Reference:**

[revised manuscript text omitted]

---

## Author Comment (AC2)

**General.**

We would like to appreciate the editor and reviewers for providing the valuable comments and a better perspective on our work to improve the manuscript. In particular, we are very grateful to the editor and reviewers for giving us the opportunity to make revision. We have revised our manuscript by fully taking the reviewers' comments into account. Responses to specific comments raised by the reviewers are described below. All the changes made and appeared in the revised text are shown in red. All detailed answers to comments are displayed in blue.

**Comments of Reviewer #2 and our responses to them**

Comments:

*This manuscript presents results from a detailed study of the chemical composition of aerosol particles collected at regular intervals over a year in Urumqi. The samples are characterized by soft ionization with UPLC-ESI-QToFMS and a focus is placed on the nitrogen containing molecular formulas identified in the mass spectra. The authors find differences in the composition of the CHON and CHN compounds between the colder and warmer periods and they attribute the majority of this difference to the variation in the fuels burned in the warmer period (wildfires) vs colder (combustion for heat). Overall, this is a very detailed and well carried out study that is clearly written. I have minor concerns about some of the data analysis and once these are addressed, I recommend acceptance of the manuscript.*

Response: We appreciate the reviewer's valuable comments on our work. Our responses to the specific comments and changes made in the manuscript are given below.

Specific Comments:

1) *Thank you for providing the data for the figures. For the peak identification, how many of the measured peaks could not be identified? Was there a mass dependence to this (i.e., were there high mass peaks that were measured that could not be identified?).*

**Response:** We greatly thank you for your professional review of our article. These are very critical issues. In this study, the number of molecules that were excluded accounted for no more than 2% of the total number of molecules. Moreover, the signal intensity of these excluded peaks also accounted for no more than 2% of the total signal intensity.

In ESI+, the identified peaks were classified into several major compound categories based on their elemental compositions, including CHO, CHON, and CHN. The CH compounds were excluded because of their small number (0.43% of the total number of compounds in ESI+) and low signal intensity ($0.33 \pm 0.28$% of the total signal intensity in ESI+) being identified in this study. In ESI−, the identified peaks were classified into CHO, CHON, CHOS, and CHONS. S-containing compounds were not discussed in this work because of our focus on N-containing compounds.

For the compounds with high mass peaks (> 700 Da), their signal intensities accounted for 1.12% and 1.37% of total signal intensities in ESI+ and ESI−, respectively. Thus, these compounds were also excluded in discussion, as indicated by many previous studies

(Wang et al., 2021; Yuan et al., 2023; Xie et al., 2020). In general, the main conclusions of this study are unaffected by the exclusion of these compounds.

More descriptions have been added in **Sect S1**, which was shown below (Pages S4-S5).

Pages S4-S5: …The CH compounds were excluded because of their small number (0.43% of the total number of compounds in ESI+) and low signal intensity ($0.33 \pm 0.28$% of the total signal intensity in ESI+) being identified in this study. For the compounds with high mass peaks (> 700 Da), their signal intensities accounted for 1.12% and 1.37% of total signal intensities in ESI+ and ESI−, respectively. Thus, these compounds were also excluded in discussion, as indicated by many previous studies (Wang et al., 2021; Yuan et al., 2023; Xie et al., 2020) …

2) *The sentence starting on line 55 is confusing and I recommend revising it: "Moreover, the modified forms of some nitrogen-containing organic compounds (NOCs) and volatile organic compounds (VOCs) by ozone (O3), hydroxyl radical (•OH), and nitrogen oxide (NOx) can lead to an increase in the health hazards of OA, among which nitrated amino acids and nitrated polycyclic aromatic hydrocarbons are two representative hazards (Franze et al., 2005; Bandowe and Meusel, 2017)." What does "modified forms" mean? The second half of the sentence (starting ...along with nitrated amino acids...) is also incomplete and may be better as its own sentence.*

**Response:** We apologize for the confusion caused by our expression and thank you for

your suggestions. The expression "modified forms" refers to the products or intermediate products of the interactions of ozone ($O_3$), hydroxyl radical (•OH), and nitrogen oxide ($NO_x$) with some nitrogen-containing organic compounds (NOCs) and volatile organic compounds (VOCs). The revision has been made in the revised manuscript (Lines 55–60).

Lines 55–60: …Moreover, the further oxidation or nitrification of some nitrogen-containing organic compounds (NOCs) and volatile organic compounds (VOCs) by ozone ($O_3$), hydroxyl radical (•OH), and nitrogen oxide ($NO_x$) can lead to an increase in the health hazards of OA (Franze et al., 2005; Bandowe and Meusel, 2017). Nitrated amino acids and nitrated PAHs are two representative hazard NOCs (Franze et al., 2005; Bandowe and Meusel, 2017).

3) On line 238, the possibility for CHO compounds to be precursors for CHON compounds is raised. Please clarify if this is referring to possible reactions in the gas-phase, in the particle-phase, or both?

**Response:** The CHON compounds can be tightly associated with secondary formation processes involving the reactions of reactive nitrogen with gas-phase and particle-phase CHO compounds (Bandowe and Meusel, 2017; Zarzana et al., 2012; Laskin et al., 2014). For example, laboratory experiments have suggested that the oxidation of isoprene and α-/β-pinene in the presence of $NO_x$ can form large amounts of CHON compounds (Surratt et al., 2010; Rollins et al., 2012; Nguyen et al., 2015). The reduced nitrogen species (e.g., $NH_3$ and $NH_4^+$) have been demonstrated to contribute to the formation of NOCs in

particle-phase (Zarzana et al., 2012; Laskin et al., 2014). Xu et al. (2023) characterized the variations of molecular compositions in urban road PM$_{2.5}$, suggesting that organic nitrates increased largely through the interactions of atmospheric oxidants, CHO compounds, and aerosol liquid water in both gas-phase and particle-phase. In general, CHO compounds can be important precursors for the formation of NOCs (via reactions in the gas- and/or particle-phases).

More discussions are presented at Lines 71–89. Based on your suggestion, the revision was made in the Lines 250–252.

Lines 250–252: …This indicated that CHO compounds may be important precursors for the formation of NOCs (via reactions in the gas- and/or particle-phases) or that they have similar origins.

4) In Table S4: how were the identifications made that are in the footnote (a, b, c, d)? For this and other tables, how is "relatively high signal" defined?

**Response:** These compounds were identified or inferred based on their MS/MS fragments or the molecular formulae of the products obtained from **Figure 5.** More descriptions were added in the page S11.

Page S11: These compounds were identified or inferred based on their MS/MS fragments or the molecular formulae of the products obtained from Figure 5.

We apologize for the confusion caused by our expression. The correct expression is that "Characteristics of the observed CHON compounds with relatively high signal intensity compared to other CHON compounds in ESI+ mode in the warm period (Page S11). The titles of other tables (**Table S6** and **Table S7**) were also updated (Pages S13-S14).

5) *The mass error calculations here look to be a little incorrect (ppm values). I agree with the assignments and the errors I calculate are within the boundaries from the paper (5 ppm), but these values should be double checked.*

**Response:** The mass errors in **Table S5** were the results directly output by the data processing software. In general, mass error calculations for mass spectrometry (Brenton and Godfrey, 2010) can be expressed as follows:

$$\Delta m_i = \frac{(m_i - m_a)}{m_a} \times 10^6 \; in \; ppm \; (parts \; per \; million)$$

where $m_i$ is the measurement value, $m_a$ is the calculated mass value.

To check the accuracy of the mass errors obtained from the software, we added the theoretical masses of the ions detected by the instrument to **Table S5** (Page S12) and calculated the mass error for each compound according to the above equation. We found that the calculated results were consistent with the output values, indicating that the mass errors obtained from the instrument are reliable.

**Once again, we deeply appreciate the time and effort you've spent in reviewing our manuscript.**

**Reference:**

Bandowe, B. A. M. and Meusel, H.: Nitrated polycyclic aromatic hydrocarbons (nitro-PAHs) in the environment – A review, Sci. Total Environ., 581-582, 237-257, https://doi.org/10.1016/j.scitotenv.2016.12.115, 2017.

Brenton, A. G. and Godfrey, A. R.: Accurate mass measurement: Terminology and treatment of data, Journal of the American Society for Mass Spectrometry, 21, 1821-1835, 10.1016/j.jasms.2010.06.006, 2010.

Franze, T., Weller, M. G., Niessner, R., and Pöschl, U.: Protein Nitration by Polluted Air, Environ. Sci. Technol., 39, 1673-1678, https://doi.org/10.1021/es0488737, 2005.

Laskin, J., Laskin, A., Nizkorodov, S. A., Roach, P., Eckert, P., Gilles, M. K., Wang, B., Lee, H. J., and Hu, Q.: Molecular Selectivity of Brown Carbon Chromophores, Environ. Sci. Technol., 48, 12047-12055, https://doi.org/10.1021/es503432r, 2014.

Nguyen, T. B., Bates, K. H., Crounse, J. D., Schwantes, R. H., Zhang, X., Kjaergaard, H. G., Surratt, J. D., Lin, P., Laskin, A., Seinfeld, J. H., and Wennberg, P. O.: Mechanism of the hydroxyl radical oxidation of methacryloyl peroxynitrate (MPAN) and its pathway toward secondary organic aerosol formation in the atmosphere, Phys. Chem.

Chem. Phys., 17, 17914-17926, https://doi.org/10.1039/C5CP02001H, 2015.

Rollins, A. W., Browne, E. C., Min, K.-E., Pusede, S. E., Wooldridge, P. J., Gentner, D. R., Goldstein, A. H., Liu, S., Day, D. A., Russell, L. M., and Cohen, R. C.: Evidence for $NO_x$ Control over Nighttime SOA Formation, Science, 337, 1210-1212, https://doi.org/10.1126/science.1221520, 2012.

Surratt, J. D., Chan, A. W. H., Eddingsaas, N. C., Chan, M., Loza, C. L., Kwan, A. J., Hersey, S. P., Flagan, R. C., Wennberg, P. O., and Seinfeld, J. H.: Reactive intermediates revealed in secondary organic aerosol formation from isoprene, P. Natl. Acad. Sci. USA, 107, 6640-6645, https://doi.org/10.1073/pnas.0911114107, 2010.

Wang, Y., Hu, M., Hu, W., Zheng, J., Niu, H., Fang, X., Xu, N., Wu, Z., Guo, S., Wu, Y., Chen, W., Lu, S., Shao, M., Xie, S., Luo, B., and Zhang, Y.: Secondary Formation of Aerosols Under Typical High-Humidity Conditions in Wintertime Sichuan Basin, China: A Contrast to the North China Plain, J. Geophys. Res.-Atmos., 126, e2021JD034560, https://doi.org/10.1029/2021JD034560, 2021.

Xie, Q., Su, S., Chen, S., Xu, Y., Cao, D., Chen, J., Ren, L., Yue, S., Zhao, W., Sun, Y., Wang, Z., Tong, H., Su, H., Cheng, Y., Kawamura, K., Jiang, G., Liu, C. Q., and Fu, P.: Molecular characterization of firework-related urban aerosols using Fourier transform ion cyclotron resonance mass spectrometry, Atmos. Chem. Phys., 20, 6803-6820,

https://doi.org/10.5194/acp-20-6803-2020, 2020.

Xu, Y., Dong, X. N., He, C., Wu, D. S., Xiao, H. W., and Xiao, H. Y.: Mist cannon trucks can exacerbate the formation of water-soluble organic aerosol and $PM_{2.5}$ pollution in the road environment, Atmos. Chem. Phys., 23, 6775-6788, https://doi.org/10.5194/acp-23-6775-2023, 2023.

Yuan, W., Huang, R.-J., Shen, J., Wang, K., Yang, L., Wang, T., Gong, Y., Cao, W., Guo, J., Ni, H., Duan, J., and Hoffmann, T.: More water-soluble brown carbon after the residential "coal-to-gas" conversion measure in urban Beijing, npj Climate and Atmospheric Science, 6, 20, 10.1038/s41612-023-00355-w, 2023.

Zarzana, K. J., De Haan, D. O., Freedman, M. A., Hasenkopf, C. A., and Tolbert, M. A.: Optical Properties of the Products of α-Dicarbonyl and Amine Reactions in Simulated Cloud Droplets, Environ. Sci. Technol., 46, 4845-4851, https://doi.org/10.1021/es2040152, 2012.

---

## Author Response (AR2)

**General.**

We would like to appreciate the editor for providing the valuable comments on our work. We have revised our manuscript by fully taking the editor's comments into account. Responses to specific comments raised by the editor are described below. All the changes made and appeared in the revised text are shown in red. All detailed answers to comments are displayed in blue.

**Comments of the Editor and our responses to them**

Comments:

*Thank you for your careful consideration of the referee comments. After careful consideration of your response document and revised manuscript, I am happy to accept your manuscript for publication following attention to a few minor corrections. Line numbers refer to the track changes version of the manuscript.*

Response: We appreciate your professional review for our article. We have revised the manuscript to address the comments. Our responses to the specific comments and changes made in the manuscript are given below.

Specific comments:

1) *Line 30: "poor-O" change to "O-poor" or "oxygen-poor"*

Response: All relevant content has been revised in the manuscript.

2) *Line 39: "Alkyl nitriles can from fresh" there is a word missing.*

Response: The revision has been made in the revised manuscript (Line 39).

Line 39: …Alkyl nitriles can be derived from fresh biomass…

3) *Line 53: Remove "however"*

Response: The revision has been made in the revised manuscript (Line 54).

4) *Line 59: "nitrogen oxide" --> "nitrogen oxides"*

Response: The revision has been made in the revised manuscript (Line 59).

5) *Line 68: This is up to the authors, but I would recommend removing the Luo et al reference as it has more of a focus on oxidative potential. This reference was added during revision.*

Response: We greatly appreciate your suggestion. The reference you mentioned has

been removed from the revised manuscript (Lines 67–68).

6) *Lines 153-154: Ozone and NOx have substantial diel variability. Please clarify how the representative values are determined.*

Response: As the $PM_{2.5}$ samples in this study were collected daily, we obtained the daily concentrations of ozone and $NO_x$ by averaging the hourly concentration data of ozone and $NO_x$ from the adjacent environmental monitoring station. We have modified this section below (Lines 150–154).

Lines 150–154: During the sampling campaigns, the meteorological data (e.g., temperature and relative humidity) and the concentrations of $O_3$ and $NO_x$ were recorded hourly from the adjacent environmental monitoring station. These hourly data were then averaged to obtain daily values to match the sampling time of $PM_{2.5}$.

7) *Sect. 2.2: Please add volumes used in the extraction processes.*

Response: We greatly appreciate your suggestions. The volumes used in the extraction processes have been added in the revised manuscript (Line 159 and Line 181).

8) *Line 165: Please define UPLC-ESI-QToFMS*

Response: The revision has been made in the revised manuscript (Lines 164–166).

Lines 164–166: …using an ultra-performance liquid chromatography quadrupole time-of-flight mass spectrometry equipped with an electrospray ionization (ESI) source (UPLC-ESI-QToFMS, Waters Acquity Xevo G2-XS)...

9) Line 350: "with poor oxygen" --> "that are oxygen poor"

Response: The revision has been made in the revised manuscript (Line 368).

10) Figure 1: Some compounds containing sulfur are included in the figure, but are not mentioned anywhere or included in the tables. I am ok with restricting the discussion to only nitrogen and CHO compounds, but in that case the sulfur containing ions should be removed from the figure. In the text (methods section perhaps), it should be clearly stated that S containing ions aren't discussed. It would be good to state the absolute number of formulas that contain S so that the reader can get a sense of how such an exclusion relates to the breadth of the dataset.

Response: We thank you for the insightful comment. The sulfur-containing ions have been removed from the **Figure 1**. The absolute number of formulas that contain S has been added in **Table S2**. Furthermore, additional statements have been added in **section 2.3**.

Lines 191–195: …primarily including CHO, CHON, and CHN groups in the ESI+ mode and CHO, CHON, CHOS and CHONS groups in the ESI– mode (Wang et al., 2017). CHOS and CHONS compounds were also detected in the ESI– mode, with

numbers of 398 and 112, respectively (**Table S2**). As this study focused mainly on NOCs, sulfur-containing species were not discussed…

11) *SI page S6 text "The identified compounds….Tong et al 2016)" or some similar version should be moved to the main text so that the reader understands how attribution to BBOA, etc. is being done. I leave it to the authors' discretion as to the best location of this text within the main manuscript.*

Response: We greatly appreciate your kind suggestion. Revisions have been made in the revised manuscript.

Lines 199–217: The identified compounds can be further classified into four subgroups based on the number of carbon atoms and $OS_C$ value (Kroll et al., 2011; Xu et al., 2023). Briefly, semi-volatile oxidized organic aerosol (SV-OOA) and low-volatility oxidized organic aerosol (LV-OOA) were associated with multi-step oxidation reactions, with $OS_C$ values between $-1$ and $+1$ and molecular formulas less than 13 carbon atoms. BBOA has $OS_C$ values ranging from $-0.5$ to $-1.5$ and more than seven carbon atmos. Compounds with $OS_C$ values less than $-1$ and carbon atoms above 20 may be related to hydrocarbon-like organic aerosol (HOA). Additionally, the modified aromaticity index ($AI_{mod}$) was also calculated to indicate the aromaticity of organic compounds (details in **Sect. S2**) (Koch and Dittmar, 2006). The van Krevelen diagrams and $AI_{mod}$ values have been proposed to further classify organic matter categories (Xu et al., 2023; Su et al., 2021), according to which the identified five subgroups included saturated-like molecules (Sa, $H/C \geq 2.0$), unsaturated aliphatic-like molecules (UA, $1.5 \leq H/C < 2.0$),

highly unsaturated-like molecules (HU, $AI_{mod} \leq 0.5$ and H/C < 1.5), highly aromatic-like molecules (HA, $0.5 < AI_{mod} \leq 0.66$), and (E) polycyclic aromatic-like molecules (PA, $AI_{mod} > 0.66$). Furthermore, it has been suggested that the above subgroups can be subdivided into O-poor and O-rich compounds depending on their O/C ratio (**Table S8**) (Merder et al., 2020; Zhong et al., 2023).

**At last, we deeply appreciate the time and effort you've spent in reviewing our manuscript.**

**Reference:**

Koch, B. P. and Dittmar, T.: From mass to structure: an aromaticity index for high-resolution mass data of natural organic matter, Rapid Commun. Mass Spectrom., 20, 926-932, https://doi.org/10.1002/rcm.2386, 2006.

Merder, J., Freund, J. A., Feudel, U., Hansen, C. T., Hawkes, J. A., Jacob, B., Klaproth, K., Niggemann, J., Noriega-Ortega, B. E., Osterholz, H., Rossel, P. E., Seidel, M., Singer, G., Stubbins, A., Waska, H., and Dittmar, T.: ICBM-OCEAN: Processing Ultrahigh-Resolution Mass Spectrometry Data of Complex Molecular Mixtures, Anal. Chem., 92, 6832-6838, https://dx.doi.org/10.1021/acs.analchem.9b05659, 2020.

Su, S., Xie, Q., Lang, Y., Cao, D., Xu, Y., Chen, J., Chen, S., Hu, W., Qi, Y., Pan, X., Sun,

Y., Wang, Z., Liu, C.-Q., Jiang, G., and Fu, P.: High Molecular Diversity of Organic Nitrogen in Urban Snow in North China, Environ. Sci. Technol., 55, 4344-4356, https://dx.doi.org/10.1021/acs.est.0c06851, 2021.

Wang, Y., Hu, M., Lin, P., Guo, Q., Wu, Z., Li, M., Zeng, L., Song, Y., Zeng, L., Wu, Y., Guo, S., Huang, X., and He, L.: Molecular Characterization of Nitrogen-Containing Organic Compounds in Humic-like Substances Emitted from Straw Residue Burning, Environ. Sci. Technol., 51, 5951-5961, https://doi.org/10.1021/acs.est.7b00248, 2017.

Xu, Y., Dong, X. N., He, C., Wu, D. S., Xiao, H. W., and Xiao, H. Y.: Mist cannon trucks can exacerbate the formation of water-soluble organic aerosol and $PM_{2.5}$ pollution in the road environment, Atmos. Chem. Phys., 23, 6775-6788, https://doi.org/10.5194/acp-23-6775-2023, 2023.

Zhong, S., Chen, S., Deng, J., Fan, Y., Zhang, Q., Xie, Q., Qi, Y., Hu, W., Wu, L., Li, X., Pavuluri, C. M., Zhu, J., Wang, X., Liu, D., Pan, X., Sun, Y., Wang, Z., Xu, Y., Tong, H., Su, H., Cheng, Y., Kawamura, K., and Fu, P.: Impact of biogenic secondary organic aerosol (SOA) loading on the molecular composition of wintertime $PM_{2.5}$ in urban Tianjin: an insight from Fourier transform ion cyclotron resonance mass spectrometry, Atmos. Chem. Phys., 23, 2061-2077, https://doi.org/10.5194/acp-23-2061-2023, 2023.